# Development of a Kinect-Based English Learning System Based on Integrating the ARCS Model with Situated Learning

**Yi-Hsing Chang [1,*], Pei-Rul Lin [1] and You-Te Lu [2]**

[1]  Department of Information Management, Southern Taiwan University of Science and Technology, Tainan 710, Taiwan; ma290204@stust.edu.tw

[2]  Department of Information and Communication, Southern Taiwan University of Science and Technology, Tainan 710, Taiwan; yowder@stust.edu.tw

*   Correspondence: yhchang@stust.edu.tw

**Abstract:** This study developed a Kinect-based somatosensory English learning system. The main design concept was to integrate Kinect as an interaction technique with theories of situated learning and the attention, relevance, confidence, and satisfaction (ARCS) model, to design relevant learning activities and materials, thereby enhancing students' learning outcomes. The proposed system allows for planning and designing learning activities and content according to situated learning components and the ARCS model. The somatosensory interaction system Kinect was used to provide users with a virtual learning environment to achieve actual spatial and physical experiences, assisting learners' engagement in stories and scenarios as well as enhancing their motivation to learn. English vocabulary related to supermarkets was set as the learning objective and 70 students ranging from third to sixth grade at a learning center in Tainan, Taiwan were selected as participants. During the experiment, participants were divided into two groups: the experimental group, which employed the proposed learning system, and the control group, in which students learned using printed materials coupled with mobile devices. Pre- and posttest scores of the two groups were used to assess learning outcomes and analyze the ARCS model-based questionnaire. The results revealed that the proposed system effectively improved learners' motivation to learn and learning outcomes.

**Keywords:** Kinect; situated learning; ARCS model; game-based learning

## 1. Introduction

Miguel [1] contended that exertion games (Exergames), such as Wii, PlayStation, and Xbox console games based on hand gesture recognition, have become a trend and are applicable to learning and rehabilitation. DePriest and Barilovits [2] and Hsu [3] indicated that physical interactions in the new generation of video games (e.g., Xbox Kinect digital games) transform conventional learning approaches. Hamzah et al. [4] introduced Kinect as an innovative learning tool that enhances teaching effects and learning experiences. Teaching through games enables learners to learn effectively and facilitates their comprehension and absorption of knowledge.

As the worldwide common language, English has always been a prevalent research topic, with numerous pertinent studies conducted. Furthermore, advancements in technology are conducive to diverse English-learning approaches and research. Katzlinger [5] and Kim [6] noted that most game-based English learning systems, such as large-scale online games and emerging video games, emphasize the enhancement of learning motivation among children. Regarding existing online and mobile English learning games, the present study also explored the learning approach of individual games and websites; some examples are listed as follows. Animal Safari [7] and Bon Appetit [8] are

simple Flash-based games, offering users simple vocabulary to learn with relatively less learning content. The website Kizclub [9] provides multiple animations and pictures. Kids Dailies [10] is a digital newspaper that offers news videos and English news reading material suitable for children. Kizclub and Kids Dailies provide relatively abundant and robust learning content centered on educational aspects of text and videos. After investigating existing games, this study discovered that they rarely provide situated learning and they lack the intuitive operation and sense of presence provided by Kinect, which invokes learners' motivation.

The attention, relevance, confidence, and satisfaction (ARCS) model of motivation was proposed by Keller [11] as a framework for encouraging students to focus on demonstrating learning motivation in learning environments. As a result, students can exhibit relatively superior learning outcomes.

Therefore, this study aimed to introduce the Kinect somatosensory interaction technique, situated learning theory, and the ARCS model of motivation into learning vocabulary and acquiring knowledge related to supermarkets. The proposed somatosensory English learning system (SELS) enables learners to engage in learning through physical interactions with virtual characters, events, and objects, thereby facilitating their motivation and learning outcomes. Accordingly, this study proposed the following research questions:

- Does the SELS improve learners' learning outcomes?
- Does the SELS enhance learners' motivation?

## 2. Literature Review

### 2.1. Situated Learning

Situated learning was proposed by Brown, Collins, and Duguid [12], who indicated that schooling content and methods provide learners with abstract and conceptual knowledge, but neglect the practice of acquired knowledge in real-life contexts. In other words, such a teaching method hinders the application of acquired knowledge in actual practice. Numerous researchers have proposed various definitions and features for components of situated learning; for instance, McLellan [13] deemed that situated learning should involve eight key components, namely stories, reflection, cognitive apprenticeship, collaboration, coaching, multiple practice, articulation of learning skills, and technology. Based on these eight key components, the present study introduced a Kinect technique into McLellan's situated learning design to combine virtual animated stories with physical locations, allowing learners to learn in realistic simulated scenarios.

### 2.2. Game-Based English Learning

Game-based learning is currently one of the most popular topics within the digital learning field. Hung et al. [14] provided a scoping overview of empirical evidence on the use and impacts of digital games in language education from 2007 to 2016. In this research, a total of 50 selected studies were systematically analyzed, and two of the revealing findings are: most of the games for language learning were custom-built by the researchers; and personal computers were the most common platforms for playing games to support language learning. On the other hand, most research concerning somatosensory-based equipment developed learning materials in conjunction with game-based learning. For example, Vukicevic et al. [15] evaluated four Kinect-based visuo-motor games called Fruits that were specially designed for this research to test whether children with autism spectrum disorder (ASD) would show behavior changes during their gameplay. The study included 10 elementary school children with ASD, aged 9–13 years, who were divided into an experimental group (n = 5) who, in addition to standard treatment, played Fruits and a control group (n = 5) who received only standard treatment during this experimental period. The preliminary findings of the research indicated a motor skill benefit for children with ASD who play Kinect-based educational games. Erman and Zeynep [16] pointed out that foreign language education is promoted in Turkey, yet because of several factors, among which weekly hours of teaching and student motivation prevail, students still cannot

develop their language skills. Therefore, they investigated the effects of game-based language learning with Kinect technology on students' self-efficacy beliefs and attitudes toward English. They developed students' communication skills by carrying out meaningful tasks based on real-life scenarios. The results revealed that there was a significant positive increase in some sub-factors of attitude and self-efficacy scores of the students.

To enhancing the learning motivation and outcome of learners, we will integrate the somatosensory control and a sense of presence into the learning activity.

### 2.3. ARCS Model of Motivation

The ARCS model of motivation encompasses the four components of attention, relevance, confidence, and satisfaction. Since this model was established, numerous researchers have applied it to learning to increase learners' motivation. For example, Fazamin et al. indicated that the ARCS model is a design framework for a learning environment in which users are encouraged to maintain their learning motivation through problem solving. Kaneko et al. [17] developed and assessed different game-based learning environments, one of which is the ARCS-based mobile platform (i.e., iOS and Android). In their study, experimental group learners were presented with games based on the ARCS model, whereas the control group was subjected to a digital learning approach. The analysis results demonstrated no significant difference in learning outcomes between the two groups; however, learners in the experimental group could concentrate more in the learning environment.

### 2.4. Kinect Application

In recent years, Kinect has been applied in numerous fields such as rehabilitation [18,19], elderly care [20,21], and digital learning. By applying Kinect to digital learning, Xu et al. [22] proposed a novel technique that helps the Kinect-based training system to select subsequential training material for users according to their real-time performance. The authors then presented an edutainment gaming system for children to illustrate the feasibility of the training method. The experimental results show that the proposed technique enhances the effect of physical training significantly. Chuang, Kuo, Fan, and Hsu [23] developed a motion-sensing digital game-based sensory integration dysfunction (SID) therapy to help the children with SID (CwSID) become more engaged in physical training. By improving their bodily-kinesthetic intelligence, these children can be more confident when facing various learning challenges. This research applied the Microsoft Kinect system and a specially designed motion-sensing game related to SID. The results were positive that the approach was able to increase the learning motivation and actions of the CwSID who participated in this study.

Based on the aforementioned studies, although Kinect is suitable for use in digital learning as a tool for human–computer interactions, the application of Kinect and other similar systems for language learning has been less well-developed.

## 3. Methodology

### 3.1. Research Architecture

The research architecture is as shown in Figure 1, containing control variable, independent variable, and dependent variable.

(1)   Control variable:

- Learning content: As supermarkets are commonly visited places, this simulated scenario enables learners to connect with the environment, thereby creating situated learning. We initially selected 30 English supermarket-related vocabulary items as the learning material from 1000 vocabulary items taught in elementary schools. They were divided into three categories: fresh produce, fruits and vegetables, and food departments. In addition,

we embedded the properties of situated learning and game-based learning into the learning content.

- Grouping method: A total of 70 students from third to sixth grade at a learning center in Tainan, Taiwan were selected as the study participants and the students were randomly divided into an experimental group and a control group.

(2) Independent variables: 35 students in the experimental group learned through the SELS, 35 students in the control group learned through conventional methods assisted with mobile devices.

(3) Dependent variables: The main purpose was to investigate the performance of the proposed system and the differences of learning achievement between the two groups.

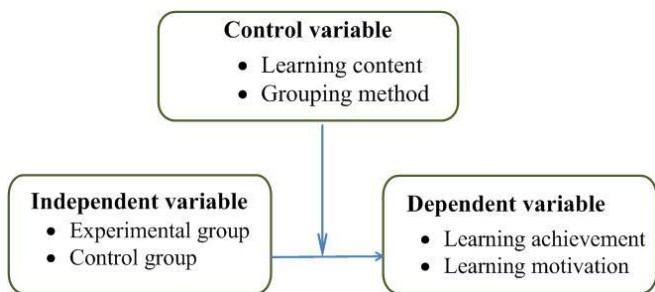

**Figure 1.** Research architecture.

*3.2. Experiment Design*

3.2.1. Experiment Duration and Procedure

The experiment duration was from June 11 to 29, 2018, and the experimental procedures were as follows:

Step 1
Experimental group: Introduced to the research for 15 min.
Control group: Introduced to the research for 15 min.
Step 2
Experimental group: A 15-min pretest was conducted before learning.
Control group: A 15-min pretest was conducted before learning.
Step 3
Experimental group: SELS learning was conducted for 60 min.
Control group: Conventional learning methods (i.e., memorizing English vocabulary from printed materials) were used for 60 min.
Step 4
Experimental group: A 15-min posttest was conducted after learning.
Control group: A 15-min posttest was conducted after learning.
Step 5
Experimental group: Completed the questionnaire for 15 min.
Control group: Interviewed for 15 min.

3.2.2. Learning Achievement

An independent samples *t*-test was used to assess the learning achievement of learners.

### 3.2.3. Learning Motivation Evaluation

This study used the following four dimensions of ARCS: attention, relevance, confidence, and satisfaction as the evaluation of learners' learning motivation. The research hypothesis consisted of four assumptions, numbered H1 to H4, as shown below. There were a total of 25 question items in the questionnaire. Questionnaire reliability was defined by Cronbach's $\alpha$, which assesses the internal consistency of a scale; $\alpha > 0.7$ implies high reliability. Respondents were asked to rate questionnaire items using a 5-point Likert scale ranging from strongly disagree (1) to strongly agree (5). The criterion value of SD was set to 1; an SD greater than 1 indicates a high variance in learners' level of agreement with the questionnaire item, whereas an SD of less than 1 implies that learners' level of agreement toward a questionnaire item was more concentrated.

**Hypothesis 1 (H1).** *There is a positive correlation between the SELS and the learning attention of learners.*

**Hypothesis 2 (H2).** *There is a positive correlation between the SELS and the learning relevance of learners.*

**Hypothesis 3 (H3).** *There is a positive correlation between the SELS and the learning confidence of learners.*

**Hypothesis 4 (H4).** *There is a positive correlation between the SELS and the learning satisfaction of learners.*

### 3.3. Integrating Situated Learning Theory and the ARCS Model of Motivation

In this section, we will explain the method of integrating the Kinect somatosensory interaction technique, situated learning theory, and the ARCS model of motivation into the proposed learning system.

### 3.3.1. Design of Situated Learning

This study employed seven of the components proposed by McLellan [12] to design its situated learning, each of which is detailed as follows:

(1) Stories: The use of virtual characters, events, and objects to combine virtual and actual environments that enabled users to immerse themselves in the scenario of a supermarket.
(2) Reflection: Educational quizzes and games provided learners with time to reflect and deliberate after learning.
(3) Cognitive apprenticeship: Learners could learn following the demonstrated static images and user manual.
(4) Coaching: In addition to the freedom of selecting their own learning content without designated objectives, learners were provided with timely reminders during learning activities.
(5) Multiple practice: Learners could repeatedly study all learning content and units.
(6) Articulation of learning skills: The activity design of a unit elucidated the learning content for the learners.
(7) Technology: Pronunciation, images, and animations in games diversified the learning process.

### 3.3.2. Design of the Learning Materials

To enhance learners' motivation, thereby increasing their learning outcomes, the design for incorporating the ARCS model into the situated learning was as follows:

(1) ARCS-A (attention): Unlike memorizing textbook vocabulary, the supermarket scenario's design attracted users, invoked their curiosity, and drew their attention.
(2) ARCS-R (relevance): The application of practical life experience allowed users to feel a sense of familiarity. Learners could set their own learning objectives, through which learning motivation was generated.

(3)　ARCS-C (confidence): Learners were provided with challenges and controllable interactions, allowing them to believe they were capable of achieving the objectives.

(4)　ARCS-S (satisfaction): Learners were provided the opportunity to apply their acquired knowledge to accomplishing games and quizzes, allowing them to display their personalities.

## 4. System Design

Based upon the design concept, the learning content was divided into the following four categories (see Figure 2). Each of the categories of fresh produce, food, and fruits and vegetables, had three parts, namely animations, vocabulary learning, and quizzes. A 3 × 3 square puzzle learning section consisted of two parts, namely quizzes and system feedback. We first explain the three parts for each of the categories as follows:

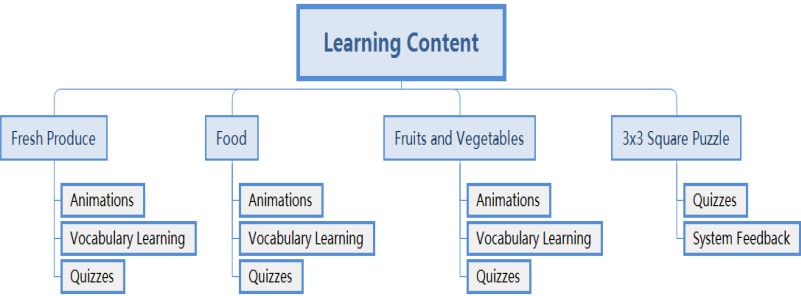

**Figure 2.** Learning materials.

### 4.1. Animation

Animation consisted of the following four parts.

(1)　Design of learners' characters: Learners march in place and their animated character should move in a corresponding manner, as shown in Figure 3. When users wish to learn or review vocabulary, they could raise either left hand or right hand, and the animated character turned to face the corresponding direction.

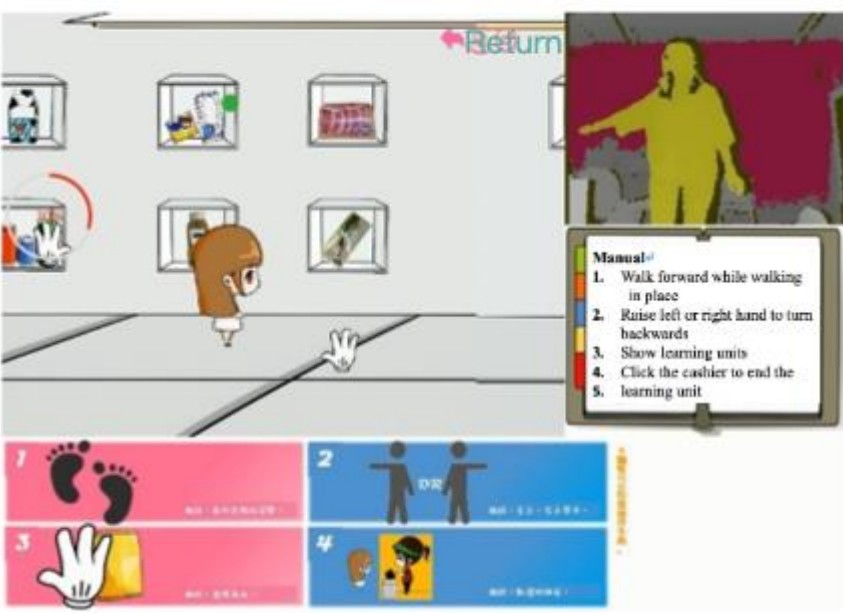

**Figure 3.** Character design.

(2)     Design of learning scenarios: Figure 4 shows the opening scene of a learning unit, which was comprised of three learning sections, namely fresh produce, fruits and vegetables, and food. The aforementioned 3 × 3 square puzzle was only available for selection after learners completed all three sections.

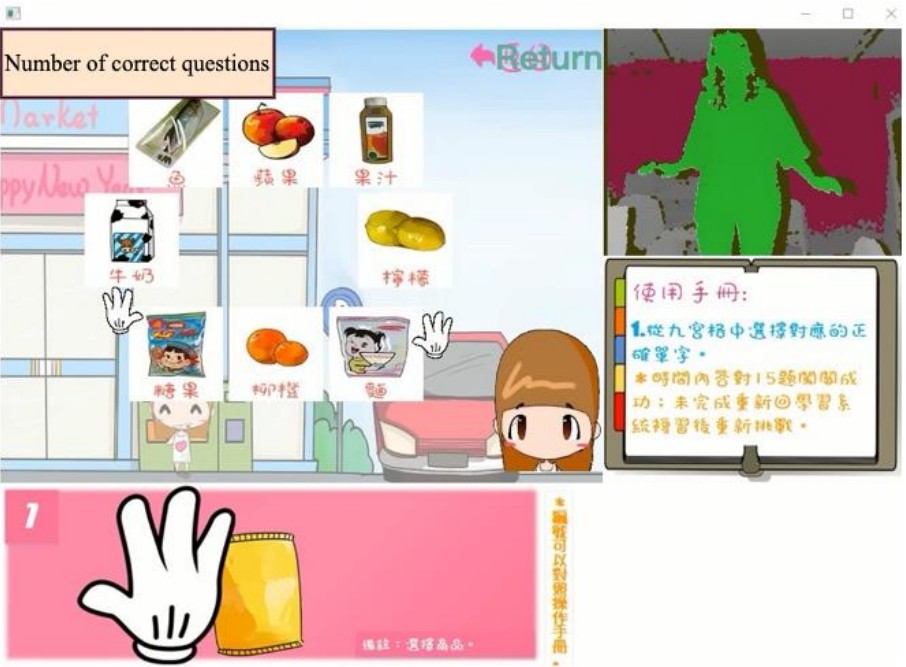

**Figure 4.** Learning scenarios.

(3)     Design of each learning section: The designated vocabulary was presented in conjunction with the corresponding picture, text, and video in each learning scenario presented through two-dimensional (2D) scrolling. Learners were provided with game instructions. After receiving text or pronunciation hints, learners could touch the image of the grocery item they wished to learn on the screen using either hand (Figure 5).

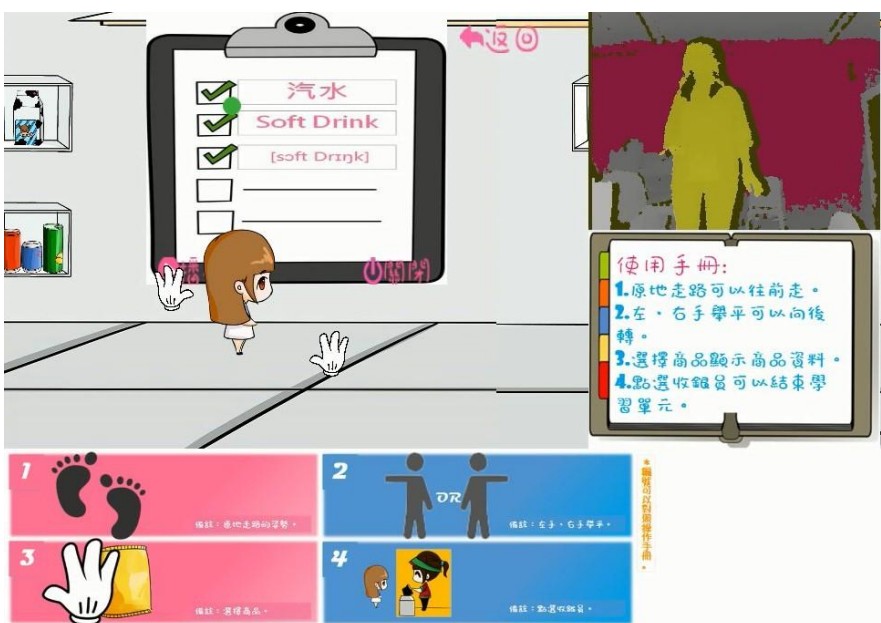

**Figure 5.** Learning section.

(4) Design of learning sections' ending scenes: When learners reached the far-right corner of the system, an animated cashier appeared. Learners could enter the somatosensory quiz on grocery-related vocabulary (Figure 6) by touching the animated cashier.

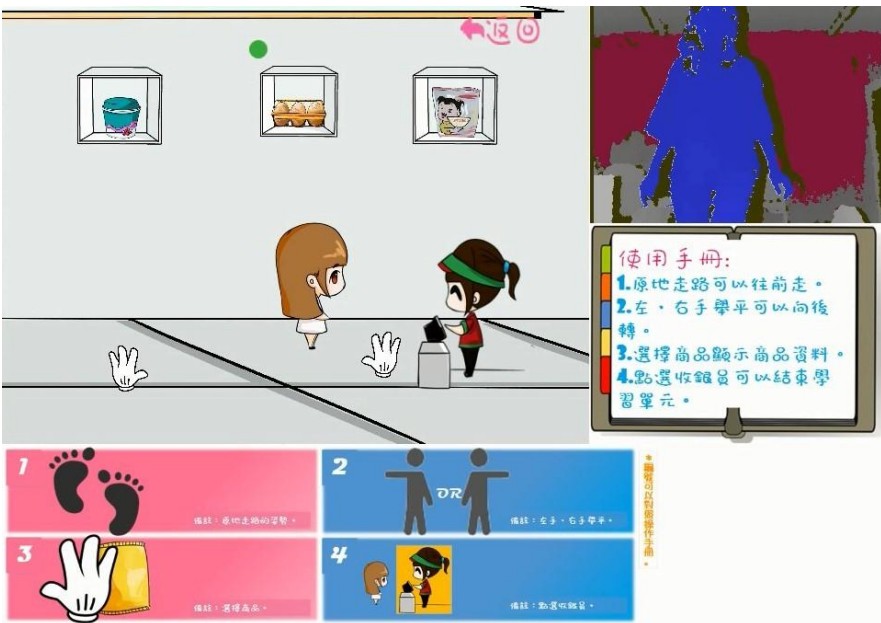

**Figure 6.** Schematic of the exit button.

### 4.2. Vocabulary Learning

The vocabulary items as the learning material were divided into three categories: fresh produce, fruits and vegetables, and food departments. Each vocabulary item was presented with a corresponding picture. When learners touched the picture with their hands, the corresponding Chinese term, English word, and pronunciation appeared. Table 1 shows the vocabulary items for the fruits and vegetables department.

**Table 1.** Vocabulary items for the fruits and vegetables department.

| Chinese | English | KK Phonetic |
|---------|---------|-------------|
| 西瓜 | Watermelon | [ˋwtmln] |
| 柳橙 | Orange | [ˋrnd] |
| 紅蘿蔔 | Carrot | [ˋkært] |
| 茄子 | Eggplant | [ˋgplænt] |
| 香蕉 | Banana | [bˋnæn] |
| 草莓 | Strawberry | [ˋstrbr] |
| 馬鈴薯 | Potato | [pˋteto] |
| 鳳梨 | Pineapple | [ˋpanæp!] |
| 蔬菜 | Vegetables | [ˋvdtb!] |
| 檸檬 | Lemon | [ˋlmn] |
| 蘋果 | Apple | [ˋæp!] |

### 4.3. Quizzes

A small quiz after each unit allowed learners to review their acquired knowledge. Five grocery items were randomly selected from a particular department. Each question displayed a picture of the grocery item along with four answer options. Learners selected the answers by touching the option button. Sound effects provided feedback to learners to indicate whether each answer was correct or

incorrect. Learners had to provide three or more correct answers to proceed to other learning units; otherwise, they had to return to the learning section to practice and retake the quiz.

In the 3 × 3 square puzzle, there were animation-based quiz questions. After accomplishing all three learning sections, learners could enter this game-based learning section to assess their learning outcomes. Learners who completed game levels using the Kinect somatosensory system gained a sense of accomplishment.

(1)   Quiz: This quiz game was presented in a 3 × 3 square puzzle format (Figure 7). A question appeared in the middle cell and the answer options appeared in the remaining eight cells for learners to select. The question automatically changed every 5 s and the system calculated the number of questions a learner answered correctly within the time limit of 1.5 min. During the quiz, learners were provided with text or pronunciation hints. Then, they could move either of their hands to touch the item button on the main screen that corresponded to the English vocabulary presented in the upper-left corner. Answering 15 or more questions correctly within the time limit indicated that a learner had successfully completed the level. The quiz question presented in Figure 7 is "Chocolate".

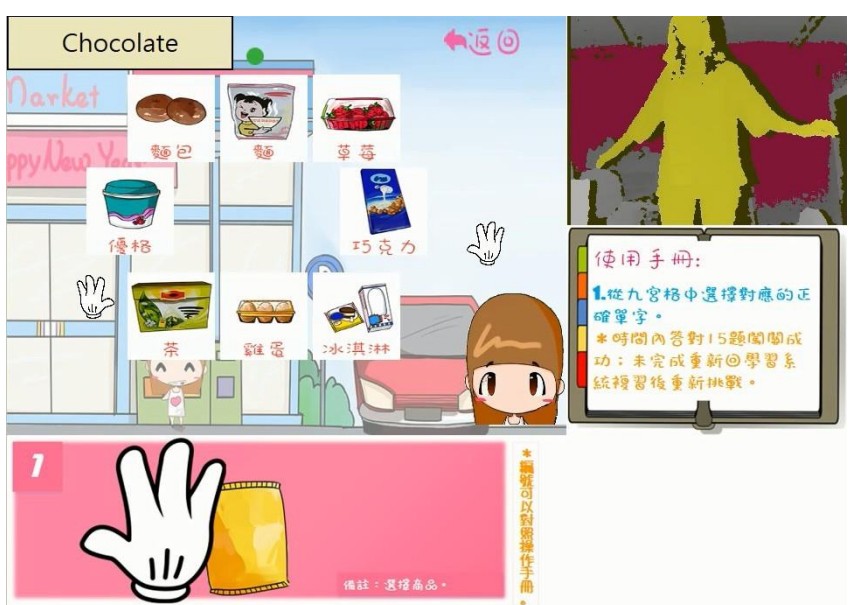

**Figure 7.** Example of SELS 3 × 3 square puzzle section.

(2)   System feedback: During the SELS 3 × 3 square puzzle game, both correct and incorrect answers triggered corresponding sound effects. Furthermore, the animated character raised signs that correspond to correct (O) or incorrect (X) answers (Figure 8).

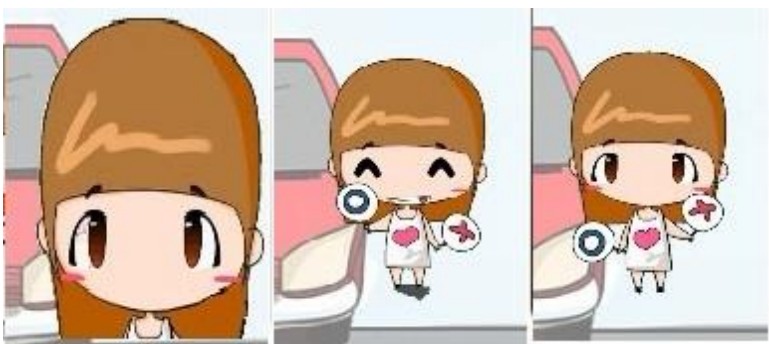

**Figure 8.** Correct and incorrect answers.

Learners who correctly answered 15 or more questions successfully completed the level, marking the end of the SELS 3 × 3 square puzzle learning activity (Figure 9).

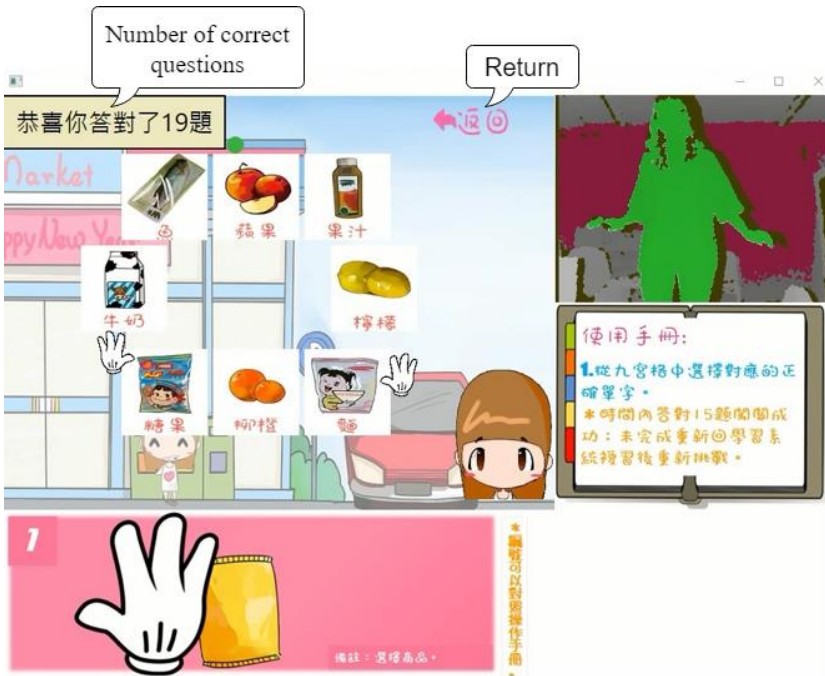

**Figure 9.** Scene showing the total questions a learner has answered correctly in the SELS 3 × 3 square puzzle.

## 5. Experimental Results

Before the start of the experiment, the prerequisite knowledge of the students in both groups was determined with regard to the learning material. An independent samples *t*-test was used to analyze the pretest results of the two groups; the results are as shown in Table 2. The mean pretest score of the experimental group was 62.429, with a standard deviation (SD) of 14.821; the mean pretest score of the control group was 62.714, with an SD of 14.519. The p-value of 0.744 did not achieve the 0.05 significance level, indicating that the pretest scores of the experimental and control groups did not differ significantly; hence, the groups' basic capability was the same.

**Table 2.** Pretest analysis (Independent sample *t*-test analysis).

|  | NO. | Mean | SD | *t* |
|---|---|---|---|---|
| Experimental | 35 | 62.429 | 14.821 | −0.329 |
| Control | 35 | 62.714 | 14.519 | |

### 5.1. Analysis of Learning Outcomes

5.1.1. Analysis of the Pre- and Posttest Results of the Experimental Group

The pre- and posttests of the experimental group were evaluated using a paired sample *t*-test, and the results are shown in Table 3. The mean of the pretest was 62.429 and that of the posttest was 76.143, leading to *p* = 0.000 (< 0.001). This statistically significant result indicated a considerable difference in the pre- and posttest scores of the experimental group.

**Table 3.** Analysis of the paired samples *t*-test of the experimental group.

| Experimental | NO. | Mean | SD | *t* |
|---|---|---|---|---|
| pretest | 35 | 62.429 | 14.821 | −12.146 |
| posttest | 35 | 76.143 | 9.000 | |

### 5.1.2. Analysis of the Pre- and Posttest Results of the Control Group

The pre- and posttests of the control group were evaluated using a paired sample *t*-test, and the results are shown in Table 4. The mean of the pretest was 62.714 and that of the posttest was 69.143. The result of $p = 0.000$ (<0.001) indicated statistical significance, and thus implied a considerable difference in the pre- and posttest scores of the control group.

**Table 4.** Analysis of the paired samples *t*-test of the control group.

| Control | NO. | Mean | SD | *t* |
|---|---|---|---|---|
| pretest | 35 | 62.714 | 14.821 | −6.349 |
| posttest | 35 | 69.143 | 11.973 | |

### 5.1.3. Posttest Analysis

In addition, an independent sample *t*-test was used to evaluate the posttest results of the experimental and control groups to determine the difference between them after learning (Table 5). The posttest score of the experimental group was 7 points higher than that of the control group, and $p = 0.000$ (<0.001) indicated statistical significance.

**Table 5.** Analysis of the independent samples *t*-tests of the experimental and control groups.

| | NO. | Mean | SD | *t* |
|---|---|---|---|---|
| Experimental | 35 | 76.143 | 9.000 | 6.803 |
| Control | 35 | 69.143 | 11.973 | |

The aforementioned analysis revealed that the mean score of the experimental group increased from 62.429 to 76.143 after learning, indicating considerable improvement. Furthermore, the control group showed significant improvement after learning, with a mean score increasing from 62.714 to 69.143. However, comparing the mean scores of the two groups after learning indicated that the experimental group had a relatively higher degree of improvement, which implies that the proposed SELS can indeed enhance learning outcomes.

### 5.2. Questionnaire Analysis

This study employed a questionnaire for qualitative analysis; 35 questionnaires were distributed, of which 35 were valid, for a valid response rate of 100%.

### 5.2.1. Reliability Analysis

Table 6 presents the analysis results. All dimensions received an $\alpha$ value higher than 0.7. The overall scale received an $\alpha$ value of 0.940, implying a certain degree of reliability.

**Table 6.** Reliability analysis of the questionnaire.

| Subscale Name | NO. of Items | Cronbach's $\alpha$ |
|---|---|---|
| ARCS-A | 5 | 0.894 |
| ARCS-R | 5 | 0.915 |
| ARCS-C | 5 | 0.899 |
| ARCS-S | 5 | 0.928 |
| Total | 20 | 0.940 |

ARCS—attention, relevance, confidence, and satisfaction; ARCS-A—ARCS-attention; ARCS-R—ARCS-relevance; ARCS-C—ARCS-confidence; ARCS-S—ARCS-satisfaction.

### 5.2.2. Analysis of Descriptive Statistics

The analysis results of ARCS-A (attention) are presented in Table 7 and indicate that all items received a mean score higher than 4, and the grand mean score was 4.506. This result shows that learners achieved a highly satisfactory level for this dimension. Therefore, hypothesis H1 is valid.

**Table 7.** Statistics of ARCS-A (attention).

| No. | Item | Mean | SD |
|---|---|---|---|
| A1 | Using the Kinect somatosensory system allows me to study harder. | 4.588 | 0.657 |
| A2 | Game scenarios motivate me to learn. | 4.647 | 0.544 |
| A3 | I can fully comprehend the teaching materials presented in the game. | 4.471 | 0.748 |
| A4 | I can fully understand the hints and instructions provided by the system. | 4.441 | 0.786 |
| A5 | I know how the game works as soon as it begins. | 4.382 | 0.817 |
| | Grand mean | 4.506 | 0.715 |

The analysis results of ARCS-R (relevance) are presented in Table 8. All questionnaire items received a mean score higher than 4, and the grand mean was 4.518; this indicated that learners achieved a highly satisfactory level for this dimension. Therefore, hypothesis H2 is valid.

**Table 8.** Statistics of ARCS-R (relevance) items.

| No. | Item | Mean | SD |
|---|---|---|---|
| R1 | Using the Kinect somatosensory system allows me to immerse myself fully in the scenarios. | 4.676 | 0.589 |
| R2 | The game makes me feel like I am actually shopping in a supermarket. | 4.500 | 0.749 |
| R3 | I can immerse myself in the game scenarios and complete the entire game. | 4.559 | 0.660 |
| R4 | I have a deeper understanding of these vocabulary items after playing this game. | 4.529 | 0.788 |
| R5 | I can think of grocery vocabulary items during the game. | 4.324 | 0.727 |
| | Grand mean | 4.518 | 0.707 |

Analysis results of ARCS-C (confidence) are shown in Table 9. All questionnaire items received a score higher than 4, and the grand mean was 4.535; this indicated that learners achieved a highly satisfactory level for this dimension. Therefore, hypothesis H3 is valid.

**Table 9.** Statistics of ARCS-C (confidence) items.

| No. | Item | Mean | SD |
|---|---|---|---|
| C1 | Using the Kinect somatosensory system increases my confidence in the learning process. | 4.588 | 0.530 |
| C2 | Using the Kinect somatosensory system is more fun than using books or mobile phones. | 4.706 | 0.524 |
| C3 | This game is highly usable. | 4.529 | 0.706 |
| C4 | This game makes me feel like I am shopping at a supermarket and allows me to learn more effectively. | 4.471 | 0.748 |
| C5 | I am confident in learning vocabulary items effectively. | 4.382 | 0.817 |
| | Grand mean | 4.535 | 0.706 |

The analysis results of ARCS-S (satisfaction) are presented in Table 10. All questionnaire items reached a score higher than 4, and the grand mean was 4.640; this indicated that learners achieved a highly satisfactory level for this dimension. Therefore, hypothesis H4 is valid.

**Table 10.** Statistics of ARCS-S (satisfaction) items.

| No. | Item | Mean | SD |
|---|---|---|---|
| S1 | In my opinion, operating the Kinect somatosensory system is convenient. | 4.382 | 0.853 |
| S2 | I am satisfied to have completed the quizzes and games. | 4.706 | 0.629 |
| S3 | This supermarket-based game is interesting. | 4.588 | 0.500 |
| S4 | I am satisfied with this system as a learning approach. | 4.559 | 0.660 |
| S5 | I would like to continue learning using this game. | 4.706 | 0.579 |
| | Grand mean | 4.640 | 0.592 |

## 6. Discussion and Conclusions

The aforementioned analysis results indicated the following findings: items A1 and A2 received the highest mean values among all items of ARCS-A. Item A2 implied that learners enjoyed the aesthetic and scenarios of the game and agree it increased their interest in learning. The elementary school students exhibited a high level of acceptance of the system's operational design and theme, both of which enabled them to concentrate on learning. Item A1 revealed that because over half of the learners were relatively inexperienced in somatosensory interfaces (controlling an animated character through body movements was a novel concept for them), they were extremely focused when using Kinect. In terms of relevance, items R1 and R3 received the highest mean scores among ARCS-R items, indicating that the learners were immersed in the supermarket scenario when learning with Kinect. That is, the integration of Kinect with situated learning stimulated students' interest in learning. Seeing these grocery items in subsequent real-world scenarios will remind the learners of the corresponding English vocabulary, which demonstrates learning outcomes. In terms of confidence, items C2 and C1 reached the highest mean scores among ARCS-C items. Item C1 implied that the adoption of Kinect enhanced learners' motivation to learn; Item C2 implied that learners preferred Kinect to books or mobile games. Therefore, the learners' confidence was boosted when using SELS, which is conducive to learning. Regarding satisfaction, items S2 and S5 received the highest mean scores among ARCS-S items, implying that learners could memorize these vocabulary items to play games. Learners acquired deep satisfaction after having successfully completed the game; therefore, the system then became acceptable to learners, generating continuance intentions to play the game. Thus, the designed game in the learning system enhanced learners' motivation to learn and increased their learning outcomes. In sum, the proposed system improved learners' motivation to learn English,

as well as strengthened their attention and concentration, enabling them to create associations between vocabulary and grocery items for effective memorization. Moreover, boosting learners' self-confidence may have facilitated their learning and created a sense of satisfaction.

On the other hand, this study had some limitations. In the part of system development, during the experiment, we discovered that the system occasionally ceased operating. Furthermore, several learners responded that the font size was too small and not clear enough, and that the descriptions were excessively long. Therefore, increasing the stability of the game and modifying the instructions could provide learners with a clearer and smoother user experience. This study employed the Kinect for Windows software development kit (SDK) 1.8 to develop the proposed system. Considering that Microsoft has released Kinect for Windows SDK 2.0 (or higher versions), subsequent developments of Kinect-based learning systems are advised to employ the latest version to benefit from abundant new features, allowing superior material design and system smoothness. In addition, this study only utilized 2D space for developing the proposed system (i.e., learners used actual body movements to interact with virtual 2D characters, events, and objects). Three-dimensional space should be developed for somatosensory learning in the future. Three-dimensional animated graphics can be integrated into the system design to cultivate a more realistic and vivid situated learning scenario, thereby allowing learners to feel as if they are learning in an actual supermarket. In the part of learning materials, nouns and verbs are equally important parts of vocabulary that competent language users need to acquire. In the study, we only targeted the learning of nouns. Therefore, in the future, we can discuss the learning effectiveness of verbs. In the part of methodology, we shall study and try some different research methodology and design such as design-based research [24] that are difficult or impossible to replicate in current practice strategies, and thus improve student learning outcomes.

In summary, the study results revealed the following findings. In terms of learning outcomes, the pre- and posttest scores of the experimental and control groups were analyzed by performing paired and independent sample *t*-tests. The obtained results indicated that the difference between the pretest scores of both groups was nonsignificant, implying that learners from both groups had similar levels of English proficiency. The result that the posttest scores of both groups were significantly different from their pretest scores shows that both somatosensory and conventional learning approaches increased learning outcomes. However, comparing the extent of progress indicated that the proposed system could more effectively improve learning outcomes. Therefore, the developed system indeed improved the learners' learning outcomes. In addition, in terms of learning motivation, the analysis results of the questionnaire showed that hypotheses H1 to H4 are all valid and indicated the SELS enhanced the learners' learning motivation.

Therefore, introducing Kinect sensor-based human–machine interactions and situated learning to learning English supermarket-related vocabulary allowed learners to immerse themselves in the scenarios through interactions with virtual characters, events, and objects, using actual physical movements. As a result, the learners became more motivated to learn and developed a desire for learning English vocabulary related to supermarkets, thereby transforming their understanding of grocery items into knowledge. Thus, the Kinect-based SELS increased learners' motivation and understanding of teaching materials, stimulating their interest in studying English vocabulary to enhance learning outcomes.

**Author Contributions:** Conceptualization, Y.-H.C.; methodology, Y.-H.C.; software, P.-R.L., and Y.-T.L.; validation, Y.-H.C., P.-R.L., and Y.-T.L.; writing—original draft preparation, Y.-H.C. and P.-R.L.; writing—review and editing, Y.-H.C. All authors have read and agreed to the published version of the manuscript.

**Funding:** This research received no external funding.

**Conflicts of Interest:** The authors declare no conflict of interest.

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
