# Peer review of "Development of a Kinect-Based English Learning System Based on Integrating the ARCS Model with Situated Learning"

_sustainability, doi:10.3390/su12052037_

Round 1

Reviewer 1 Report

Results: including an analysis of the relationship between the experience developed and gender would be interesting (linear regression)

Author Response

Thank you for your valuable and important comments. We re-organize the manuscript as following:

  1. In Section 3, we add two subsections, 3.1 research architecture and 3.2 Experiment Design. In addtion, the original subsection 3.1 Design of Situated Learning and 3.2 Design of the Learning Materials are combined into section 3. Integrating situated learning theory and the ARCS model of motivation.
  2. The original section 3.2 Design of the Learning Materials to become a new section, Section 4 System Design.

Reviewer 2 Report

An interesting article that explores an under-researched area of gamification using somatosensory control to enhance English Language Learning. It is however let down by the choice of a control-group quasi experimental research design methodology that results in the usual ‘no-significant difference’ in leader outcomes.

An English grammar check is needed throughout.

Wider interaction with the literature on Gamification?

Edmonds, Roger, & Smith, Simon. (2016, 24-26 October). Location-based mobile learning games: Motivation for and engagement with the learning process. Paper presented at the Mobile Learning Futures – Sustaining Quality Research and Practice in Mobile Learning, Proceedings of the 15th World Conference on Mobile and Contextual Learning, mLearn 2016, UTS, Sydney, Australia.

Clarke & Arnab https://www.researchgate.net/project/Gamechangers

p1 line 38 - the sentence needs a more defined subject: ‘English’ should be ‘English language learning’ or similar

Why was the context of Supermarkets chosen for the study? There should be a discussion justifying the choice of this particular learning context. Section (2) Vocabulary learning should precede the other sections, and become section (1) of the learning design methodology.

The design methodology section should show a more explicit link to the stated underlying learning theories: situated learning and game-based learning.

The use of a control group methodology is problematic - As Reeves argues (McKenney, Susan, & Reeves, Thomas. (2019). Conducting educational design research (2nd ed.). London: Routledge.

Reeves, Thomas. (2015). Educational design research: Signs of progress. Australasian Journal of Educational Technology, 31(5), 613-620. doi: http://dx.doi.org/10.14742/ajet.2902

Reeves, Thomas. (2005). No significant differences revisited: A historical perspective on the research informing contemporary online learning. In G. Kearsley (Ed.), Online learning: Personal reflections on the transformation of education (pp. 299-308). Englewood Cliffs, NJ: Educational Technology Publications.

), comparative educational research methodologies invariably result in ‘no-significant difference’ in learning outcomes, a better methodology for designing learning interventions is therefore Design-Based research, that allows the design of learning outcomes that are difficult or impossible to replicate in current practice strategies, and thus improve student learning outcomes.

Author Response

Thank you for your valuable and important comments. We re-organize the manuscript as following:

  1. In Section 3, we add two subsections, 3.1 research architecture and 3.2 Experiment Design. In addtion, the original subsection 3.1 Design of Situated Learning and 3.2 Design of the Learning Materials are combined into section 3. Integrating situated learning theory and the ARCS model of motivation.
  2. The original section 3.2 Design of the Learning Materials to become a new section, Section 4 System Design.

Comment 1:

  • An interesting article that explores an under-researched area of gamification using somatosensory control to enhance English Language Learning. It is however let down by the choice of a control-group quasi experimental research design methodology that results in the usual ‘no-significant difference’ in leader outcomes.
  • The use of a control group methodology is problematic - As Reeves argues (McKenney, Susan, & Reeves, Thomas. (2019). Conducting educational design research(2nd ed.). London: Routledge.
  • Reeves, Thomas. (2015). Educational design research: Signs of progress. Australasian Journal of Educational Technology, 31(5), 613-620. doi: http://dx.doi.org/10.14742/ajet.2902
  • Reeves, Thomas. (2005). No significant differences revisited: A historical perspective on the research informing contemporary online learning. In G. Kearsley (Ed.), Online learning: Personal reflections on the transformation of education(pp. 299-308). Englewood Cliffs, NJ: Educational Technology Publications.
  • ), comparative educational research methodologies invariably result in ‘no-significant difference’ in learning outcomes, a better methodology for designing learning interventions is therefore Design-Based research, that allows the design of learning outcomes that are difficult or impossible to replicate in current practice strategies, and thus improve student learning outcomes.

Response: Thanks the reviewer provide the important information. We will consider the “Design-Based research” in future. (lines 392- 395)

Comment 2: An English grammar check is needed throughout.

Response: the manuscript is translated from Chinese to English by Wallace, https://www.editing.tw/

Comment 3: p1 line 38 - the sentence needs a more defined subject: ‘English’ should be ‘English language learning’ or similar

Response: changed

 Comment 4: Why was the context of Supermarkets chosen for the study? There should be a discussion justifying the choice of this particular learning context. Section (2) Vocabulary learning should precede the other sections, and become section (1) of the learning design methodology.

Response: re-organize the manuscript.

 Comment 5:The design methodology section should show a more explicit link to the stated underlying learning theories: situated learning and game-based learning.

 Response: re-organize the manuscript and explain in the learning content (Figure 1).

Reviewer 3 Report

The introduction and literature review can be improved. I don’t think it is fair to conclude that situated learning is neglected from the review unless you can quote published reviews. MAll games are designed with scenarios to situate learning. Please argue specifically which part of situated learning are you talking about? Perhaps along the line of embedded bodily?

It may be better to change the last sentence of 2.2 “however, they lack somatosensory control, a sense of presence..” to “however, integration of somatosensory control, a sense of presence… could further promote the learning of English.

For section 2.4, it seems good to point out that the application of Kinect and other similar systems for language learning has been less well developed.   

The description of the control group activities sounds like paper-based traditional teaching rather than aided with mobile devices. Please clarify.

The findings are clear and convincing.

The diagrams contain only Chinese characters. A bilingual presentation could be better.  

The review of existing games and this study seems to point to the exclusive attention to the learning of nouns but verbs are equally important parts of vocabulary that competent language users need to acquire. Kinect may promote the learning of verbs. The authors may suggest this for future research.

Author Response

Thank you for your valuable and important comments. We re-organize the manuscript as following:

  1. In Section 3, we add two subsections, 3.1 research architecture and 3.2 Experiment Design. In addtion, the original subsection 3.1 Design of Situated Learning and 3.2 Design of the Learning Materials are combined into section 3. Integrating situated learning theory and the ARCS model of motivation.
  2. The original section 3.2 Design of the Learning Materials to become a new section, Section 4 System Design.

Comment 1: The introduction and literature review can be improved. I don’t think it is fair to conclude that situated learning is neglected from the review unless you can quote published reviews. MAll games are designed with scenarios to situate learning. Please argue specifically which part of situated learning are you talking about? Perhaps along the line of embedded bodily?

Response: explain in section 3.3.1

Comment 2: It may be better to change the last sentence of 2.2 “however, they lack somatosensory control, a sense of presence..” to “however, integration of somatosensory control, a sense of presence… could further promote the learning of English.

Response: Changed.

Comment 3: For section 2.4, it seems good to point out that the application of Kinect and other similar systems for language learning has been less well developed.  

Response: Thanks. We added into the manuscript. 

Comment 4: The description of the control group activities sounds like paper-based traditional teaching rather than aided with mobile devices. Please clarify.

Response: the control group can use the mobile devices to search the related data.

Comment 5: The diagrams contain only Chinese characters. A bilingual presentation could be better.  

Response: Figures 3, 4, and 9 were modified to be bilingual presentation.

Comment 6: The review of existing games and this study seems to point to the exclusive attention to the learning of nouns but verbs are equally important parts of vocabulary that competent language users need to acquire. Kinect may promote the learning of verbs. The authors may suggest this for future research.

Response: We already add the comments into the future work (lines 389 to 395) .

Round 2

Reviewer 2 Report

The authors have responded positively to the reviewers comments and the article is much improved.

A minor revision needed: The McKenney and Reeves reference in the References Section is incorrect, it should be:

McKenney, S., & Reeves, T. (2019). Conducting educational design research (2nd ed.). London: Routledge.

Author Response

updated. Thanks.